# Intratumoral Delivery of Interleukin 9 via Oncolytic Vaccinia Virus Elicits Potent Antitumor Effects in Tumor Models

**DOI:** 10.3390/cancers16051021

**Published:** 2024-02-29

**Authors:** Junjie Ye, Lingjuan Chen, Julia Waltermire, Jinshun Zhao, Jinghua Ren, Zongsheng Guo, David L. Bartlett, Zuqiang Liu

**Affiliations:** 1Allegheny Health Network Cancer Institute, Pittsburgh, PA 15212, USA; yejunjie@whu.edu.cn (J.Y.); chenlingjuan@hust.edu.cn (L.C.); jwaltermire@hotmail.com (J.W.); jinshunzhao1960@gmail.com (J.Z.); 2Department of Surgery, Drexel University College of Medicine, Philadelphia, PA 19104, USA; 3Department of Cancer Center, Renmin Hospital of Wuhan University, Wuhan 430060, China; 4Cancer Center, Union Hospital, Tongji Medical College, Huazhong University of Science and Technology, Wuhan 430022, China; jhrenmed@hust.edu.cn; 5Department of Immunology, Roswell Park Comprehensive Cancer Center, Buffalo, NY 14203, USA; zongsheng.guo@roswellpark.org

**Keywords:** oncolytic virus, colon cancer, IL-9, IL-10, tumor microenvironment, modulation, anti-CTLA-4 antibody, immune checkpoint blockade, combination, immunotherapy

## Abstract

**Simple Summary:**

The success of cancer immunotherapy is largely associated with immunologically hot tumors. Oncolytic viruses can transform cold tumors into hot tumors. In this study, we found that an oncolytic vaccinia virus (oVV) expressing interleukin-9 (IL-9) can transform cold tumors into hot tumors and that it elicited effective antitumor effects. The antitumor effects can be enhanced in combination with an immune checkpoint blockade, indicating a potential translation of the IL-9-expressing oncolytic virus into a clinical trial to enhance the antitumor effects elicited by immune checkpoint blockades for cancer immunotherapy.

**Abstract:**

The success of cancer immunotherapy is largely associated with immunologically hot tumors. Approaches that promote the infiltration of immune cells into tumor beds are urgently needed to transform cold tumors into hot tumors. Oncolytic viruses can transform the tumor microenvironment (TME), resulting in immunologically hot tumors. Cytokines are good candidates for arming oncolytic viruses to enhance their function in this transformation. Here, we used the oncolytic vaccinia virus (oVV) to deliver interleukin-9 (IL-9) into the tumor bed and explored its antitumor effects in colon and lung tumor models. Our data show that IL-9 prolongs viral persistence, which is probably mediated by the up-regulation of IL-10. The vvDD-IL-9 treatment elevated the expression of Th1 chemokines and antitumor factors such as IFN-γ, granzyme B, and perforin. IL-9 expression increased the percentages of CD4^+^ and CD8^+^ T cells in the TME and decreased the percentage of oVV-induced immune suppressive myeloid-derived suppressor cells (MDSC), leading to potent antitumor effects compared with parental virus treatment. The vvDD-IL-9 treatment also increased the percentage of regulatory T cells (Tregs) in the TME and elevated the expression of immune checkpoint molecules such as PD-1, PD-L1, and CTLA-4, but not GITR. The combination therapy of vvDD-IL-9 and the anti-CTLA-4 antibody, but not the anti-GITR antibody, induced systemic tumor-specific antitumor immunity and significantly extended the overall survival of mice, indicating a potential translation of the IL-9-expressing oncolytic virus into a clinical trial to enhance the antitumor effects elicited by an immune checkpoint blockade for cancer immunotherapy.

## 1. Introduction

Cancer immunotherapy has joined surgery, chemotherapy, and radiotherapy as a successful modality to treat cancers. Immunologically hot or ‘T-cell-inflamed’ tumors are more responsive to immunotherapy approaches. Unfortunately, the majority of solid tumors are defined as cold or immune-desert tumors [1,2]. Therefore, there is an urgent need to develop new approaches that can improve T cell infiltration and tip the cancer-immune set point in the TME [3], resulting in immunologically hot tumors, so as to improve the therapeutic efficacy and application of cancer immunotherapy. 

Oncolytic viruses (OVs) can directly kill tumor cells, and this killing can release tumor-associated antigens (TAAs), neoantigens, damage-associated molecular pattern (DAMP) molecules, OV-derived pathogen-associated molecular pattern (PAMP) molecules, and inflammatory cytokines to trigger innate and adaptive antitumor immunity. This antitumor immune reactivity results in the infiltration of diverse immune cells, including T lymphocytes, into the TME [4,5]. OVs have positive effects on almost every aspect of the cancer-immunity cycle and can be further armed with chemokines, cytokines, or other molecules to modulate the TME so as to harness the immune system to attack and treat tumors [6,7,8,9,10,11]. Oncolytic viral therapy can be considered a form of immunotherapy and has been suggested to be the next remarkable trend in cancer immunotherapy [12,13]. 

T cells play a pivotal role in antitumor immunity. CD8^+^ T cells can directly kill cancer cells [14], while CD4^+^ T cells usually combat cancers indirectly. CD4^+^ T cells can differentiate into various subsets, mainly including Th1 (Type 1 T helper cells), Th2, Th9, Th17, and regulatory T cells (Tregs). In general, Th1 cells can secret Th1 cytokines such as IFN-γ and IL-2 to enhance the antitumor efficacy of CD8^+^ T cells and NK cells, and Tregs can suppress the antitumor function of T cells [15] and NK cells [16]; while the function of Th2 and Th17 cells in tumor immunity are inconsistently reported to be either anti-tumoral [17,18,19] or pro-tumoral [20,21,22]. 

Th9 cells were named for their high levels of cytokine IL-9 secretion [23,24]. IL-9 is a T cell growth factor [25] and was also characterized to be essential for murine lymphoma growth [26]. IL-9 has been shown to promote the development of multiple human liquid tumors, including nasal natural killer (NK)/T-cell lymphoma [27], anaplastic large-cell lymphoma [28], and B-cell non-Hodgkin’s lymphoma [29], and the IL-9 antibody can inhibit the proliferation of these cells. On the other side, IL-9 has also been shown to have antitumor roles in the development of melanoma [19,30], lung carcinoma [30], and colon cancers [31]. IL-9 antibody treatment promoted B16 melanoma lung metastasis [19], while recombinant IL-9 treatment inhibited B16 melanoma and Lewis lung cancer growth in mice. In addition, increased B16 melanoma growth was observed in IL-9 receptor knockout mice [30]. The overexpression of IL-9 in a murine colon cancer CT26 model led to the transformation of the tumor microenvironment and delayed tumor growth [31]. IL-9 was recently demonstrated to inhibit the metastatic potential of breast and cervical cancers via controlling extracellular matrix remodeling and cellular contractility [32]. To our knowledge, no studies have been published using OVs to deliver IL-9 for cancer treatment. In the current study, we asked whether and how the intratumoral expression of IL-9 using an oVV modulates the TME and what antitumor effects it might have in murine tumor models. 

## 2. Material and Methods

### 2.1. Mice and Cell Lines

Female C57BL/6 (B6 in short) and BALB/c mice approximately 5–6 weeks old were purchased from The Jackson Laboratory (Bar Harbor, ME, USA) and housed in specific pathogen-free conditions in the Allegheny Health Network Research Institute Preclinical Facility. All animal studies were approved by Allegheny Health Network Research Institute Institutional Animal Care and Use Committee. Mouse colon cancer MC38-luc, CT26-luc, and mesothelioma AB12-luc cells were generated by the infection of parental tumor cells with firefly luciferase-carrying lentivirus and antibiotic blasticidin selection. African green monkey kidney fibroblast CV1, human embryonic kidney 293 (HEK293), HeLa, mouse melanoma B16, and Lewis lung cancer (LLC) cells were obtained from American Type Culture Collection (Manassas, VA, USA). HEK293 and HeLa cells were grown in Dulbecco’s Modified Eagle’s medium (DMEM) supplemented with 20% calf bovine serum (CBS), 2 mM L-glutamine, and 1 × penicillin/streptomycin in a 37 °C, 5% CO_2_ incubator. Other cell lines were grown in DMEM supplemented with 10% fetal bovine serum (FBS), 2 mM L-glutamine, and 1 × penicillin/streptomycin in a 37 °C, 5% CO_2_ incubator (PHCbi, Wood Dale, IL, USA).

### 2.2. Virus Generation

VSC20, a *vgf* gene-inactivated Western Reserve strain vaccinia virus, was used as a parental virus for homologous recombination. Murine IL-9 cDNA was amplified from pGEM-IL-9 (Sino Biological, Wayne, PA, USA) using PCR (Forward primer: GGCGGTCGACATGTTGGTGACATACATCC; Reverse primer: CCGCGGCGCGCCTCATGGTCGGCTTTTCTGCC). The cDNA fragment was digested with SalI + AscI and ligated via T4 DNA ligase into a shuttle plasmid pCMS1-IRES-YFP, resulting in a new shuttle plasmid pCMS1-IL-9-YFP. In this shuttle vector, IL-9 gene expression is driven by a vaccinia virus promoter pSe/l, and selection marker YFP is expressed from another vaccinia virus promoter p7.5 in the new shuttle plasmid. The shuttle vector was used for homologous recombination of murine *IL-9* plus *yfp* marker into the *tk* locus of the vaccinia viral genome of VSC20. To make the new virus, vvDD-IL-9, CV-1 cells were infected with VSC20 at a multiplicity of infection (MOI) of 0.1 for 2 h and then transfected with the shuttle plasmid pCMS1-IL-9-YFP. The progeny viruses were harvested 3 days later and used to infect new CV1 cells for the selection of the new recombinant virus vvDD-IL-9 using fluorescence-activated cell sorting based on the expression of yellow fluorescent protein 24 h after infection. A double viral gene-inactivated (*tk-* and *vgf-*) vaccinia virus carrying *yfp* cDNA at its *tk* locus, vvDD-YFP (vvDD in short), was the control virus for this work.

### 2.3. Viral Replication and IL-9 Expression In Vitro

MC38-luc (3.0 × 10^5^/well), AB12-luc (3.0 × 10^5^/well), or CT26-luc (3.0 × 10^5^/well) cells were seeded in 24-well culture plates on day 0. On day 1, the seeded cells were infected with vvDD or vvDD-IL-9 at an MOI of 1 in 0.15 mL of 2% FBS-containing DMEM per well for 2 h and 0.35 mL of 10% FBS containing-DMEM was added into each well. The cells were cultured and harvested at 24 h after infection. The culture supernatants were harvested to measure IL-9 using ELISA (enzyme-linked immunosorbent assay) (BioLegend, San Jose, CA, USA) and the cell pellets were applied to extract RNA to measure the viral gene A34R to monitor viral replication and transgene IL-9 expression by RT-qPCR, respectively. 

### 2.4. Cytotoxicity Assay In Vitro

Tumor cells were plated at 1.0 × 10^4^ (except B16 cells, which were plated at 5.0 × 10^3^) cells per well in 96-well culture plates and infected with indicated viruses the next day at different MOIs. Cell viability was determined at 48 h after infection using Cell Counting Kit-8 (Boster Biological Technology, Pleasanton, CA, USA), and was performed according to the manufacturer’s instructions.

### 2.5. Rodent Tumor Models

B6 mice were intraperitoneally (i.p.) inoculated with 5.0 × 10^5^ MC38-luc cancer cells and divided into required groups on the indicated day after tumor cell inoculation according to tumor size based on live-animal IVIS imaging performed using a Xenogen IVIS 200 Optical In Vivo Imaging System (Caliper Life Sciences, Hopkinton, MA, USA). Grouped mice were i.p. injected with indicated viruses or PBS. In some experiments, anti-CTLA-4 Ab (clone 9D9; Ichorbio: #ICH1096; 200 µg/injection) and anti-GITR Ab (clone DTA-1, Bio X Cell: #BE0063; 200 µg/injection) were i.p. injected into mice alone or in combination with indicated viruses. In some experiments, mice were euthanized to harvest peritoneal tumor nodules for further analysis. 

B6 mice were subcutaneously (s.c.) inoculated with 5.0 × 10^5^ LLC cells into the right flanks. Tumor-bearing mice were intratumorally (i.t.) injected with 60 µL PBS or 5.0 × 10^7^ PFU (plaque forming units) or 60 µL virus per mouse 9 days after tumor cell inoculation. MC38-luc-tumor-bearing B6 mice receiving the combination therapy of vvDD-IL-9 and anti-CTLA-4 Ab, which had survived for more than 120 days and were confirmed to be tumor-free via live-animal IVIS imaging, were s.c. inoculated with 1.0 × 10^6^ MC38 and 5.0 × 10^5^ B16 tumor cells into their bilateral flanks, respectively. Naïve B6 mice received the same treatments as controls. Tumor size was measured using an electric caliper in two perpendicular diameters.

### 2.6. Flow Cytometry

Collected tumor tissues were weighed and incubated in RPMI 1640 medium containing 2% FBS, 1 mg/mL collagenase IV (Sigma, St. Louis, MO, USA: #C5138), 0.1 mg hyaluronidase (Sigma, St. Louis, MO, USA: #H6254), and 200 U DNase I (Sigma, St. Louis, MO, USA: #D5025) at 37 °C for 1–2 h to make single cells. In vitro virus-infected cells or single cells from tumor tissues were blocked with α-CD16/32 Ab (eBioscience, San Diego, CA, USA: clone 93; #14-0161-85; 1:1000) and then stained with antibodies against mouse CD45 (BioLegend, San Diego, CA, USA: #103132; PerCP-Cy5.5, clone: 30-F11, 1:300), CD4 (Brilliant Violet 421, BioLegend, San Diego, CA, USA: clone: GK1.5, #100438; 1:300), Foxp3 (BioLegend, San Diego, CA, USA: #118904; PE, clone: QA20A67, 1:100), CD8 (BioLegend, San Diego, CA, USA: #100714; APC-Cy7, clone: 53-6.7, 1:300), IL-9 receptor (BioLegend, San Diego, CA, USA: #158806; APC, clone: S18011D, 1:300), CCR6 (BioLegend, San Diego, CA, USA: #129815; PE-Cy7, clone: 29-2L17, 1:300), CD11b (BioLegend, San Diego, CA, USA #101216; PE-Cy7, clone: M1/70), and Gr-1 (BioLegend, San Diego, CA, USA: #108406; FITC, clone: RB6-8C5, 1:300). The intracellular staining kit for Foxp3 staining was purchased from BioLegend. Samples were collected on MACSQuant^®^ Analyzer 10 Flow Cytometer (Miltenyi Biotec., Auburn, CA, USA) and the data were analyzed using FlowJo software v10 (Tree Star Inc., Ashland, OR, USA).

### 2.7. RT-qPCR

Total RNA was extracted from virus-infected cells or tumor tissues using RNeasy Kit (Qiagen, Valencia, CA, USA). One microgram of RNA was used for cDNA synthesis, and 25 to 50 ng of subsequent cDNA was used to conduct mRNA expression TaqMan analysis or SYBR Green analysis on the Quantagene q225 qPCR System (Kubo Technology Co., Ltd., Beijing, China). All primers for TaqMan analysis were purchased from Thermo Fisher Scientific (Waltham, MA, USA). The primers for glucocorticoid-induced tumor necrosis factor receptor family-related protein (GITR) (p1: GCATATGTGTCACACCTGAGTA; p2: CCGGAAGCCAAACACAATATC) and HPRT1 (p3: GGATACAGGCCAGACTTTGTT; p4: ACGTGATTCAAATCCCTGAAGTA) used in SYBR Green analysis were synthesized by Azenta, Inc. (Burlington, MA, USA). Gene expression was normalized to the housekeeping gene HPRT1 and expressed as fold increase (2^−ΔCT^), where ΔCT = CT _(Target gene)_ − CT _(HPRT1)_.

### 2.8. Statistics

Statistical analyses were performed using unpaired Student’s *t* test (GraphPad Prism version 9). Data are means ± SD. Animal survival is presented using Kaplan–Meier survival curves and was statistically analyzed using a log-rank test (GraphPad Prism version 9). Tumor growth cures were statistically analyzed using two-way ANOVA (GraphPad Prism version 9). Values of *p* < 0.05 were considered statistically significant, and all *p* values were two-sided. In the figures, standard symbols are used: * *p* < 0.05; ** *p* < 0.01; *** *p* < 0.001; and **** *p* < 0.0001. ns: not significant.

## 3. Results

### 3.1. IL-9 Expression Does Not Impact Viral Replication and Cytotoxicity In Vitro

We used vvDD, a double viral gene-deficient (*tk*- and *vgf*-) oVV, to express murine IL-9, and this newly constructed virus was called vvDD-IL-9. MC38-luc cells were infected with vvDD-IL-9 or the control virus (vvDD) at an MOI of 1. Twenty-four hours post viral infection, total RNA from virus-infected MC38-luc cells were isolated and applied for RT-qPCR assay. The results showed that the viral gene (A34R) mRNA levels were similar between these two viruses, while IL-9 mRNA levels were significantly higher in vvDD-IL-9-treated MC38-luc cells. A similar pattern was found in the AB12-luc and CT26-luc cancer cell lines receiving the same treatment (Figure 1A).

We further measured the amount of IL-9 in the cell culture supernatants by ELISA. The mean amounts of IL-9 in the cell culture supernatants from vvDD-IL-9-infected MC38-luc, AB12-luc, or CT26-luc cells were 34.38 ng/mL, 85.93 ng/mL, or 72.19 ng/mL, respectively (Figure 1B). These data showed that transgene IL-9 was successfully expressed and secreted from infection cells via oVV delivery. To investigate the possible impact of IL-9 expression on virus cytotoxicity in vitro, MC38-luc cells were infected with vvDD-IL-9 or the control virus vvDD at MOIs of 0, 0.05, 0.1, 0.5, 1, or 5, respectively, and the cell viability was measured 48 h after infection. The results showed that the cell viability was similar and dose-dependent after the infection with vvDD-IL-9 or vvDD (Figure 2A). 

A similar pattern was found in the other three cancer cells (AB-12-luc, B16, and CT-26-luc) with the same treatment (Figure 2B–D). The data suggest that IL-9 expression has no additional impact on the viral cytotoxicity capacity of vvDD-IL-9. 

### 3.2. IL-9 Expressing oVV Elicits Antitumor Effects in Tumor Models

To evaluate the antitumor efficacy of vvDD-IL-9, B6 mice were i.p. injected with MC38-luc colon cancer cells and these tumor-bearing mice were i.p. injected with vvDD or vvDD-IL-9 at a dose of 2.0 × 10^8^ PFU/200 µL per mouse or 200 µL PBS (mock treatment) 5 days after tumor injection. The dates of the spontaneous death or euthanasia of treated mice meeting the experimental endpoints were recorded. The virus vvDD-IL-9 elicited significantly more potent antitumor effects compared to PBS or vvDD treatment as shown by extending the survival of the tumor-bearing mice (Figure 3A).

The median survival time of mice receiving vvDD-IL-9 treatment was 43 days, while the median survival time of mice receiving vvDD or PBS treatment was 31 or 18 days, respectively. We also explored the therapeutic efficacy of vvDD-IL-9 using a subcutaneous tumor model. B6 mice were s.c. inoculated with 1.0 × 10^6^ Lewis lung cancer (LLC) cells into their right flanks and primary tumors were intratumorally (i.t.) injected with PBS, vvDD, or vvDD-IL-9 when the tumor volumes were 50–60 mm^3^, typically about 9 days after tumor cell injection. The virus vvDD-IL-9 treatment significantly retarded tumor growth compared to the treatments with PBS (vvDD-IL-9 vs. PBS, *p* < 0.0001 at day 30 after treatment) or vvDD (vvDD-IL-9 vs. vvDD, *p* = 0.0002 at day 30 after treatment) (Figure 3B). All treated mice were euthanized to count the number of lung metastatic tumor nodules when the first mouse with a tumor whose size was over 2 cm in any direction was found. The results showed that both viral treatments significantly reduced the lung metastases, and the vvDD-IL-9 treatment worked slightly better in preventing metastases, though this difference was not significant when compared to the vvDD treatment (Figure 3C). These data demonstrated that a potent therapeutic effect was elicited by vvDD-IL-9. 

### 3.3. IL-9 Expressing oVV Modulates the Tumor Microenvironment

To explore how vvDD-IL-9 elicits antitumor effects, we investigated the TME of MC38-luc after treatment. B6 mice were i.p. injected with MC38-luc tumor cells and 5 days later were i.p. injected with PBS, vvDD, or vvDD-IL-9, respectively. Treated mice were euthanized and tumor nodules were collected on day 5 after treatment. Single cells were made from tumor nodules and analyzed using flow cytometry. The results showed that the numbers of tumor-infiltrating CD4^+^ and CD8^+^ T cells were significantly elevated after viral treatment, compared with PBS treatment. vvDD-IL-9 treatment significantly increased the number of tumor-infiltrating CD4^+^ T cells and CD8^+^ T cells, compared to parental virus vvDD treatment (Figure 4A,B). 

Interestingly, the CCR6^+^CD8^+^ T cells were significantly elevated after the vvDD-IL-9 treatment compared to vvDD treatment (Figure 4C). More IL-9 receptor-expressing tumor-infiltrating leukocytes were observed after the vvDD-IL-9 treatment compared to the vvDD treatment (Figure 4D), indicating that the IL-9/IL-9 receptor signal pathways are involved in the antitumor effects elicited by vvDD-IL-9 treatment. We also found that immunosuppressive myeloid-derived suppressor cells (MDSCs, CD11b^+^Gr-1^+^) were slightly decreased after the vvDD-IL-9 treatment when compared to the vvDD treatment, while vvDD, but not vvDD-IL-9, significantly increased the percentage of MDSCs in tumors when compared to the PBS treatment (Figure 4E) [33]. The mean percentage of the tumoral MDSCs after PBS, vvDD and vvDD-IL-9 treatment is 0.78%, 3.1%, and 1.5%, respectively. Immunosuppressive Tregs (CD4^+^Foxp3^+^) were significantly increased after vvDD-IL-9 treatment (Figure 4F), which might weaken the antitumor effects elicited by the vvDD-IL-9 treatment.

We further investigated the dynamic modulation of the TME after vvDD-IL-9 treatment. The treated mice were euthanized on days 5 and 9, and tumor nodules were collected and total RNA was extracted for RT-qPCR analysis. We first measured the viral transgene expression. The vvDD-IL-9 treatment significantly elevated the mRNA of the viral gene A34R at both days 5 and 9 compared to the vvDD treatment, indicating that IL-9 might improve the viral replication and/or survival of the virus in vivo (Figure 5A).

As expected, the vvDD-IL-9 treatment also led to a significant increase in the mRNA of IL-9 at both days 5 and 9 compared to the vvDD treatment (Figure 5B). Since the immunosuppressive cytokine IL-10-expressing oVV was reported to have a prolonged viral persistence [34], we measured IL-10 mRNA levels in the tumor nodules. On day 5, the level of IL-10 mRNA was similar after both virus treatments. However, the vvDD-IL-9 treatment significantly elevated the levels of IL-10 mRNA in the tumor on day 9 (Figure 5C), suggesting that secondary IL-10 upregulation might enhance the persistence of vvDD-IL-9 in the tumor. The mRNA of the IL-9 receptor in tumors was significantly increased on day 9, but not on day 5 (Figure 5D). It was reported that the CCL20/CCR6 pathway is involved in the IL-9-mediated leukocyte infiltration into tumors [19], so we measured the mRNA levels of CCL20 and CCR6 in tumors. The results showed that the mRNA levels of CCL20 and CCR6 were significantly elevated on day 9, though not on day 5, after the vvDD-IL-9 treatment compared to the vvDD treatment (Figure 5E,F). We further measured the expression of Th1 chemokines and other mediators associated with anti-/pro- tumor immunity in the TME. On day 5 after treatment, the vvDD-IL-9 treatment induced a similar expression of the Th1 chemokine CXCL9 compared with the vvDD treatment; however, the vvDD-IL-9 treatment induced a significantly higher CXCL9 expression on day 9 after treatment compared to the vvDD treatment (Figure 5G). A similar pattern was found for the expression of two other Th1 chemokines: CXCL10 (Figure 5H) and CXCL11 (Figure 5I). The antitumor-immunity-associated factors IFN-γ, granzyme B (GzmB), and perforin had similar expression patterns as Th1 cytokines (Figure 5J–L). The expression pattern of immune checkpoint molecules PD-1, PD-L1 and CTLA-4 were similar on day 5 after both viral treatments, but the vvDD-IL-9 treatment induced a significantly bigger increase in these immune checkpoint molecules on day 9 after treatment (Figure 6A–C) compared to the vvDD treatment, which might also weaken the antitumor effects induced by the vvDD-IL-9 treatment. 

### 3.4. CTLA-4 Blockade Enhanced the Antitumor Effects Elicited by vvDD-IL-9

As described above, vvDD-IL-9 treatment can elicit potent antitumor effects, but can also induce more Tregs and a higher level of immune checkpoint molecule CTLA-4 expression. Considering that CTLA-4 can mediate the immunosuppressive function of Tregs [35] and the anti-GITR antibody (clone DTA-1) can deplete the number of Tregs which express high levels of GITR [36], we asked whether the antitumor effects elicited by the vvDD-IL-9 treatment can be enhanced by the blockade of either CTLA-4 or GITR. B6 mice were injected i.p. with MC38-luc cells and these tumor-bearing mice were treated 5 days after tumor cell inoculation (Figure 6D). The mice which received the combination therapy of vvDD-IL-9 and anti-CTLA-4 antibody survived significantly longer than the mice which received monotherapies of vvDD-IL-9 or anti-CTLA-4 antibody or the combination therapy of vvDD and anti-CTLA-4 antibody. The median survival time of the mice receiving combination therapy of vvDD-IL-9 and anti-CTLA-4 antibody was 60 days, while the median survival time of mice receiving vvDD-IL-9 or the combination of vvDD and anti-CTLA-4 antibody was 43 or 45 days, respectively, suggesting a non-redundant role of the CTLA-4 blockade and virally delivered IL-9 leading to improved antitumor effects being elicited by the combination therapy (Figure 6E). It is interesting that the combination of vvDD-IL-9 and anti-GITR antibody did extend the overall survival when compared to vvDD and the anti-GITR antibody but did not when compared to the vvDD-IL-9 treatment, suggesting that the anti-GITR antibody had a limited role in the combination therapy (Figure 6E). We tried to seek the reason for the failure of the combination therapy of vvDD-IL-9 and anti-GITR antibody. We investigated the GITR mRNA expression in tumors via RT-qPCR analysis. The data showed that the GITR expression in tumor was significantly down-regulated after viral treatments with either vvDD or vvDD-IL-9 compared to PBS, whereas there was no significant difference between the two viral treatments. This might explain the failure of the DTA-1 treatment in the current experiment (Figure 6F). The MC38-luc tumor-bearing mice which were cured by the combination therapy of vvDD-IL-9 and anti-CTLA-4 antibody were re-challenged with a high dose of MC38 (1.0 × 10^6^ per mouse) in the left flank and challenged with irrelevant tumor control B16 (5.0 × 10^5^ per mouse) in the right flank. Naïve B6 mice were used as controls. Both tumors grew well in the naïve B6 mice, indicating the good transplantability of both tumor cell lines. All MC38-luc tumor-cured mice rejected the re-challenge with MC38 but not that with the irrelevant tumor cells B16 (Figure 6G,H), suggesting an established systemic tumor-specific antitumor immunity in these mice. Collectively, these data demonstrate that the combination therapy of vvDD-IL-9 and the anti-CTLA-4 antibody, but not the anti-GITR antibody, elicited potent systemic tumor-specific antitumor effects in a murine colon cancer model.

## 4. Discussion

Cancer immunotherapy has been proven to be a successful modality to treat cancer. However, its responsiveness is mostly limited to hot, inflamed tumors. As most human solid tumors are cold [1,37], new approaches are urgently needed to improve T cell infiltration and tip the cancer-immune set point in the TME [3] to improve the therapeutic efficacy and application of cancer immunotherapy. Replication-competent OVs can transform cold tumors into hot tumors and exert antitumor responses beyond their oncolytic nature, especially when these viruses are armed with genes such as cytokines, chemokines, and costimulatory molecules in order to augment antitumor immunity [4,10,11,38,39,40,41,42,43,44,45,46,47,48]. OVs can be applied as cancer immunotherapy agents either alone or in combination with other cancer therapies [6,38,39,40,42,43,44,45,46,47,48,49,50,51,52,53,54,55,56].

The parental OV, vvDD, with the inactivation of both the thymidine kinase (TK) and vaccinia growth factor (VGF) genes, has been proven to be safe in phase 1 clinical trials [57,58,59]. The intratumoral expression of the chemokine CXCL11, cytokine IL-15Ra, or IL-23 using vvDD has been shown to efficiently transform the TME and induce potent antitumor effects [8,42,60]. Membrane-bound IL-2- or IL-12-expressing vvDD avoided the severe toxicity associated with systemic exposure while modulating the TME and induced superior antitumor effects, especially in combination with the PD-1 blockade, curing all or most of the mice with a high tumor burden [48]. In this study, IL-9 was delivered into tumor beds via vvDD and strong antitumor effects have been demonstrated in two non/low immunogenic tumor models. To explore the mechanisms of the antitumor effects elicited by vvDD-IL-9, we first investigated the TME using a murine colon cancer model. Our data showed that vvDD-IL-9 treatment significantly increased CD4^+^ and CD8^+^ T cells in the tumors, compared with vvDD treatment. Our data also showed that CCR6^+^CD8^+^ T cells with the characteristics of early effector memory cells [61] were increased in tumor after vvDD-IL-9 treatment, suggesting that IL-9 might promote CCL20/CCR6-dependent recruitment of CCR6^+^CD8^+^ T cells into tumor beds, in agreement with the phenomena previously described in a lung metastatic B16 melanoma model [19]. The virus vvDD, but not vvDD-IL-9, promoted MDSC accumulation in tumors, suggesting that IL-9 might down-regulate MDSC accumulation since MDSC was demonstrated to rapidly accumulate at the site of vaccinia virus infection [33]. We then investigated the key immune profile in the TME using RT-qPCR. We found that both viral replication and IL-9 expression are significantly elevated after vvDD-IL-9 treatment at either day 5 or day 9 after treatment. IL-10 expression in tumors receiving vvDD-IL-9 treatment was significantly elevated at day 9 after treatment, compared with vvDD treatment, indicating a possible role of IL-10 in vvDD-IL-9 persistence. Previously, an IL-10-expressing oVV was reported to achieve extended viral persistence [34]. Consistent with CCR6^+^CD8^+^ T cell accumulation in tumors, the expression of CCL20 and CCR6 in tumor were significantly increased at day 9 after vvDD-IL-9 treatment, compared with vvDD treatment, suggesting a role of CCL20/CCR6-dependent T cell recruitment in the antitumor effects elicited by vvDD-IL-9 treatment. It is interesting that vvDD-IL-9 receptor expression in tumors was significantly different at day 9. In fact, IL-9 receptor has been reported to be expressed on non-hematopoietic cells such as epithelial cells of the respiratory system and smooth muscle cells [62]. The vvDD-IL-9 treatment further elevated Th1 chemokines including CXCL9, CXCL10 and CXCL11, and antitumor factors such as IFN-γ, granzyme B and perforin. The vvDD-IL-9 treatment increased infiltrating T cells (both CD8^+^ and CD4^+^), down-regulated MDSC, together with the elevated intratumoral expression of Th1 chemokines and antitumor factors, leading to the transformation of an immune-suppressive TME to an immune-favorable one, and potent antitumor effects.

Tregs were significantly increased after vvDD-IL-9 treatment, compared with vvDD treatment, and IL-9 has previously been reported to enhance the suppressive function of Tregs [29,63], which might weaken the antitumor effects elicited by vvDD-IL-9 treatment. As CTLA-4 plays a pivotal role in the immunosuppressive function of Tregs [35] and the intratumoral expression of CTLA-4 was significantly elevated after vvDD-IL-9 treatment, we asked whether a CTLA-4 blockade can enhance the antitumor effects elicited by vvDD-IL-9 treatment. GITR is highly expressed on Tregs and the anti-GITR antibody (clone DTA-1) can deplete Tregs in vivo [36]. We also asked whether DTA-1 treatment can enhance the antitumor effects elicited by vvDD-IL-9 treatment. Our data demonstrated that the anti-CTLA-4 antibody, but not the anti-GITR antibody DTA-1, can induce significantly improved tumor-specific antitumor effects in combination with vvDD-IL-9 treatment. The dramatically down-regulated GITR expression after viral treatment might be a possible reason for the failure of the combination therapy of vvDD-IL-9 and the DTA-1 antibody. 

There are a number of limitations with the OV in our current study, some of which are shared with other OVs. First, IL-9 can act as a double-edged sword in cancer development [64]. IL-9 always plays an antitumor function in most solid tumors while it promotes tumor progression in hematological neoplasms and in some solid tumors. Therefore, caution must be exercised when targeting an IL-9-armed OV towards a specific type of cancer. Our study has used murine models representing colon and lung cancer. Before its translation into a human clinical trial, more careful studies with multiple tumor models are needed in which it is to be translated into human patients with certain types of cancer. Second, OVs, especially when armed with an interleukin, can usually increase immune checkpoint molecule expression in tumors. These immunosuppressive effects can be reduced with an antibody blockade at immune checkpoints. We have recently shown that immune checkpoint molecules such as PD-1/PD-L1 are significantly upgraded after vaccinia virus treatment, and the combination treatment of a vaccinia virus and an anti-PD-1/PD-L1 antibody can elicit superior antitumor effects [42,44,48,50,51]. In the current study, vvDD-IL-9 treatment also enhanced the expression of PD-1/PD-L1 in addition to CTLA-4. We have shown that a CTLA-4 blockade can enhance vvDD-IL-9-elicited antitumor effects, but we have not investigated the combination treatment of vvDD-IL-9 and a PD-1/PD-L1 blockade. Prior to the clinical translation of these findings, it is worth investigating the antitumor effects elicited by a combination of vvDD-IL-9 and a PD-1/PD-L1 blockade, or vvDD-IL-9 and the blockade of PD-1/PD-L1 plus CTLA-4. OVs can be delivered by intratumoral, intraperitoneal, and intravenous injections. The vaccinia virus JX-594, which is TK-inactivated and expresses human GM-CSF, has been effective at a high dose delivered by intravenous injection [65]. vvDD has previously been demonstrated to be safe for both intravenous and intratumoral injection [58,59]. For the further development of the systemic delivery of an OV, the PI3Kδ inhibitor was reported to transiently inhibit macrophage activities and promote the efficiency of the intravenous delivery of an oncolytic virus to the tumor [66]. This approach deserves further exploration in combination with our novel OV in preclinical studies and potentially applies to cancer patients in clinical trials. 

The clinical translation of our cytokine-armed OV is challenging due to the rapid immune clearance of the virus and the hurdle of infecting tumors throughout the body using a systemic injection. While intratumoral injection has limited utility in managing metastatic tumors, using the local viral infection to induce a systemic immune response is possible [50]. We and others have reported in situ therapeutic cancer vaccination strategies using a local injection of OV in combination with systemic immune agents. A local injection of OV can also be used as a pre-treatment to enhance tumor infiltrating lymphocyte harvesting for adoptive cell therapy [67]. Many other translational opportunities exist, including regional delivery for the treatment of peritoneal or pleural based metastatic tumors [68,69,70]. 

## 5. Conclusions

In summary, our current study demonstrates that vvDD-IL-9 treatment can deliver IL-9 into the tumor bed and elevate IL-10 expression. The accumulated IL-10 may, in turn, prolong viral persistence and enhance IL-9 accumulation. The viral-induced immunogenic cell death and IL-9 accumulation might work together to transform the immunosuppressive TME and elicit potent antitumor effects in colon and lung cancer models. The combination therapy of an anti-CTLA-4 antibody and vvDD-IL-9 significantly improves overall survival. In conclusion, an IL-9-expressing OV alone or in combination with anti-CTLA-4 antibody should be considered for a clinical trial.

## Figures and Tables

**Figure 1 cancers-16-01021-f001:**
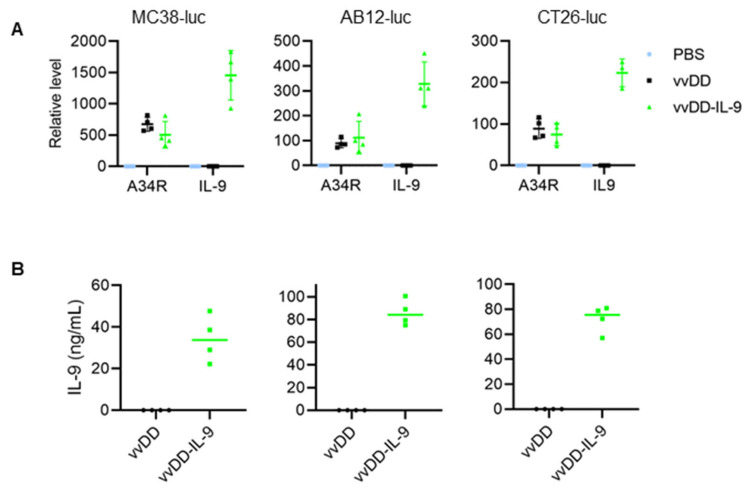
vvDD-IL-9 infection shows significantly higher IL-9 secretion and a similar viral replication compared vvDD in vitro. Tumor cells of MC38-luc (3.0 × 10^5^ cells), AB12-luc (3.0 × 10^5^ cells), or CT26-luc (3.0 × 10^5^ cells) were mock-infected or infected with vvDD or vvDD-IL-9 at an MOI of 1. At 24 h post infection, the cells were pelleted and harvested to measure A34R or IL-9 expression using RT-qPCR (**A**) and the supernatants were harvested 24 h after infection to measure IL-9 amount using ELISA (**B**), respectively.

**Figure 2 cancers-16-01021-f002:**
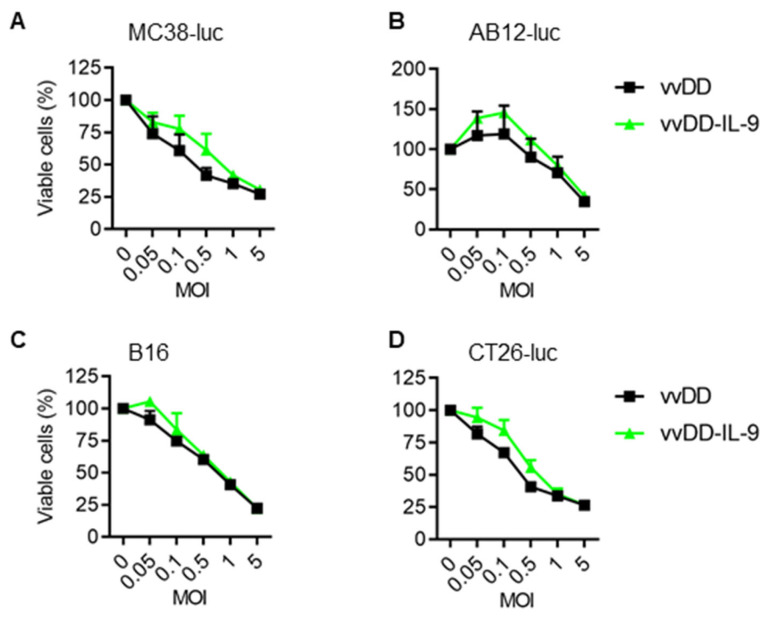
vvDD-IL-9 has a similar viral cytotoxicity compared with vvDD in vitro. Tumor cells of MC38-luc (1.0 × 10^4^ cells) (**A**), AB12-luc (8.0 × 10^3^ cells) (**B**), B16 (8.0 × 10^3^ cells) (**C**), or CT-26-luc (8.0 × 10^3^ cells) (**D**), were infected with vvDD-IL-9 or vvDD at indicated MOIs. Cell viability was measured 48 h after viral infection.

**Figure 3 cancers-16-01021-f003:**
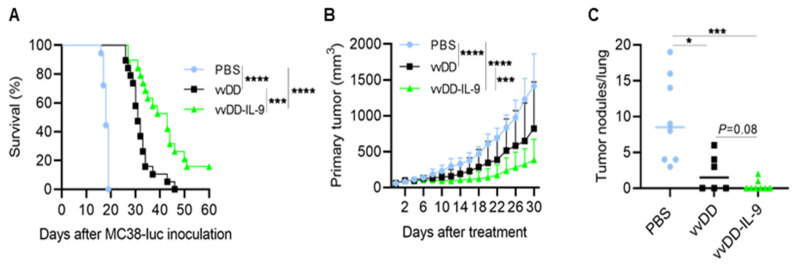
vvDD-IL-9 treatment elicits potent therapeutic effects in murine tumor models. B6 mice were i.p. inoculated with 5.0 × 10^5^ MC38-luc cells and then treated with PBS, vvDD, or vvDD-IL-9 at 2.0 × 10^8^ PFU/mouse five days after tumor cell inoculation. The Kaplan–Meier survival curve is shown (**A**). B6 mice were s.c. inoculated with 1.0 × 10^6^ LLC cells in right flank and were i.t. treated with 60 µL PBS or 60 µL virus (5.0 × 10^7^ PFU) per mouse on day 9 after tumor cell inoculation. The primary tumor size was measured every two days, and all treated mice were euthanized after the first mouse with a tumor size over 2 cm in any direction was found to count lung metastatic tumor nodules. Tumor growth curves (**B**) and lung metastatic tumor nodules (**C**) are shown, respectively. A log-rank (Mantel–Cox) test was used to compare survival rates. A two-way ANOVA test was used to compare tumor growth cures. *: *p* < 0.05; ***: *p* < 0.001; and ****: *p* < 0.0001.

**Figure 4 cancers-16-01021-f004:**
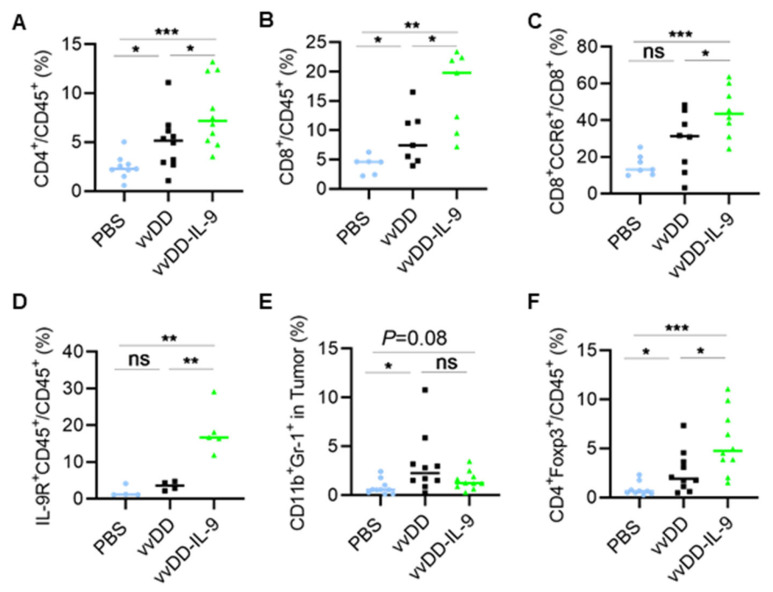
vvDD-IL-9 treatment transforms tumor-infiltrating leukocytes. B6 mice were i.p. inoculated with 5.0 × 10^5^ MC38-luc cells and treated with PBS, vvDD, or vvDD-IL-9 at 2.0 × 10^8^ PFU/mouse five days after tumor cell inoculation. Tumor-bearing mice were euthanized five days after treatment and primary tumors were collected. Tumor infiltrating leukocytes were analyzed using flow cytometry. The percentage of CD4^+^ T cells (**A**), CD8^+^ T cells (**B**), CCR6^+^CD8^+^ T cells (**C**), IL-9R+CD45+ cells (**D**), MDSCs (**E**) and Tregs (**F**) is shown, respectively. * *p* < 0.05; ** *p* < 0.01; *** *p* < 0.001; ns: not significant.

**Figure 5 cancers-16-01021-f005:**
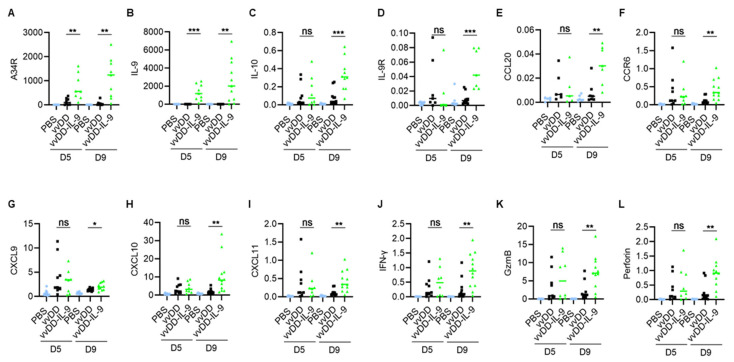
vvDD-IL-9 treatment transforms tumor immunity-associated factors in TME. B6 mice were i.p. inoculated with 5.0 × 10^5^ MC38-luc cells and treated with PBS, vvDD, or vvDD-IL-9 at 2.0 × 10^8^ PFU/mouse five days after tumor cell inoculation. Tumor-bearing mice were euthanized to collect primary tumors five days or nine days after treatment. The expression of factors (**A**–**L**) associated with tumor effects in the TME was analyzed using RT-qPCR. * *p* < 0.05; ** *p* < 0.01 and *** *p* < 0.001. ns: not significant.

**Figure 6 cancers-16-01021-f006:**
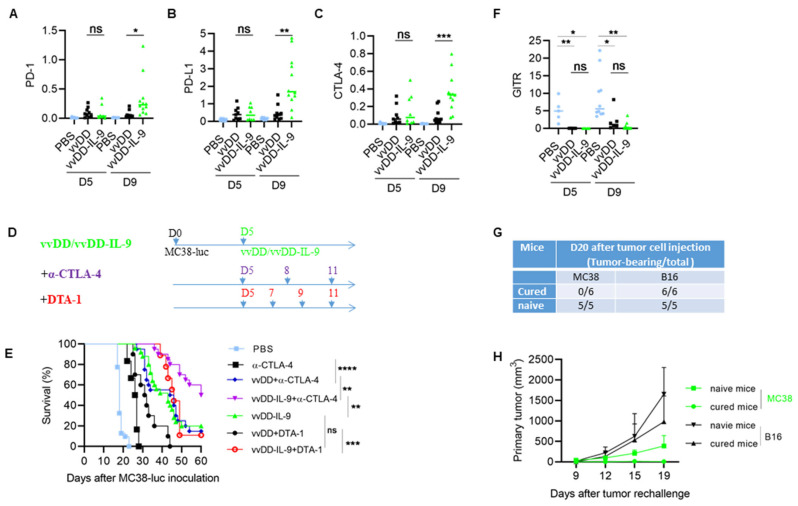
vvDD-IL-9 treatment elicits superior antitumor effects in the combination of anti-CTLA-4 antibody, but not DTA-1 antibody. B6 mice were i.p. inoculated with 5.0 × 10^5^ MC38-luc cells and, 5 days later, treated with PBS, vvDD, or vvDD-IL-9 at 2.0 × 10^8^ PFU/mouse. Tumor-bearing mice were euthanized to collect primary tumors five days or nine days after treatment. The expression of immune checkpoint molecules PD-1, PD-L1, CTLA-4 in tumors was analyzed using RT-qPCR (**A**–**C**). B6 mice were i.p. inoculated with 5.0 × 10^5^ MC38-luc and treated with PBS, anti-CTLA-4 antibody (200 µg/injection), vvDD (2.0 × 10^8^ PFU/mouse) plus anti-CTLA-4 antibody (200 µg/injection), vvDD-IL-9 (2.0 × 10^8^ PFU/mouse) plus anti-CTLA-4 antibody (200 µg/injection), vvDD (2.0 × 10^8^ PFU/mouse) plus DTA-1 antibody (200 µg/injection), vvDD-IL-9 (2.0 × 10^8^ PFU/mouse) plus DTA-1 antibody (200 µg/injection), or vvDD-IL-9 at 2.0 × 10^8^ PFU/mouse five days after tumor cell inoculation (**D**) and the survival curves are shown (**E**). B6 mice were i.p. inoculated with 5.0 × 10^5^ MC38-luc cells and treated with PBS, vvDD, or vvDD-IL-9 at 2.0 × 10^8^ PFU/mouse five days after tumor cell inoculation. Tumor-bearing mice were euthanized to collect primary tumors five days or nine days after treatment. The expression of immune checkpoint molecule GITR was analyzed using RT-qPCR. (**F**). Naïve B6 mice or MC38-luc-tumor-bearing B6 mice treated with vvDD-IL-9 plus anti-CTLA-4 antibody mentioned above, which had survived for more than 120 days, were s.c. challenged with MC38 tumor cells (1.0 × 10^6^) in the left flanks and irrelative B16 tumor cells (5.0 × 10^5^) in the right flanks. The tumor-bearing mouse ratio is shown in (**G**) and the tumor growth curve is shown in (**H**). * *p* < 0.05; ** *p* < 0.01; *** *p* < 0.001; and **** *p* < 0.0001; ns: not significant.

## Data Availability

All data are available from the corresponding authors upon reasonable request.

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
