# Peer review of "Intratumoral Delivery of Interleukin 9 via Oncolytic Vaccinia Virus Elicits Potent Antitumor Effects in Tumor Models"

_cancers, 2024, doi:10.3390/cancers16051021_

Round 1
Reviewer 1 Report
Comments and Suggestions for Authors
Review of manuscript “Intratumoral Delivery of Interleukin 9 via Oncolytic Vaccinia Virus Elicits Potent Antitumor Effects in Tumor Models” by Junjie Ye et al..
In the present study the authors make use of a newly devised oncolytic vaccinia virus (named vvDD-IL-9) for additional expression of interleukin-9 (IL-9) in different rodent models for colon carcinoma, melanoma and lung cancer. As compared to the parental virus, expression of IL-9 prolonged persistence of the oncolytic virus, most probably mediated by up-regulation of IL-10. The nicely presented data clearly show that the vvDD-IL-9 treatment elevated the expression of a series of Th1 chemokines, and anti-tumor factors leading to the establishment of a potentially “T-cell-inflamed“ tumor. Whereas the vvDD-IL-9 treatment also increased the percentage of Tregs in the TME and elevated the expression of immune checkpoint molecules such as PD-1, PD-L1 and CTLA-4, a combination therapy of vvDD-IL-9 and anti-CTLA-4 antibody could counteract most of theses effects. The overall concept of the study is clearly visible. However, the initial ratio for using IL-9 is not quite evident. As the authors themselves note, IL-9 has been demonstrated to possess both anti-tumoral and pro-tumoral effects and other interleukins like IL-2 or IL-12 mostly lacking anti-tumoral effects have already been shown to be quite effective in tumor treatment. The authors should address this point in more detail in the discussion section of a revised version of the manuscript (see also major point 1 below).
Major points:
1) The discussion section for large parts is merely a repetition of the results section. The authors should put a greater emphasis on comparing their novel oncolytic vaccinia virus with other systems employing either oncolytic vaccinia viruses or viruses in general for tumor treatment. Especially the aspect of translation into the clinic should be addressed in more detail, thereby also considering the proposed combination of an oncolytic virus with antibodies for neutralization of possible immunsuppressive effects mediated by the interleukine transgene expression of the oncolytic virus.
Minor points:
1) Abstract line 27: abbreviation MDSC (myeloid-derived suppressor cells) should be explained.
2) Introduction line 78: Please insert “the“ to change to “In the current study“.
3) Materials and Methods line 103: typo, omit “a“ between shuttle and plasmid.
4) Materials and Methods line 103 ff.: I strongly assume that in the shuttle plasmid pCMS1-IL-9-YFP the interleukin-9 is expressed from a CMV promoter and the YFP is expressed via internal translation from the IRES. This should be described more clearly.
5) Line 144: typo, please omit “were“.
Comments on the Quality of English Language
Minor editing of English language required.
Reviewer 2 Report
Comments and Suggestions for Authors
The article explores the use of oncolytic vaccinia virus to deliver Interleukin 9 intratumorally. The IL-9-armed oncolytic vaccinia virus induces immune responses and alters the tumour microenvironment in mouse models. When combined with anti-CTLA-4 antibody treatment, the Interleukin 9-expressing oncolytic virus induces systemic tumour-specific antitumour immunity, extending overall survival.
These mechanisms collectively demonstrate the ability of Interleukin 9 delivered via oncolytic vaccinia virus to modulate the tumour microenvironment, enhance immune responses, and improve the efficacy of cancer immunotherapy. The study indicates a potential for the use of IL-9-armed oncolytic virus to improve cancer immunotherapy outcomes. Overall, the study was well conducted, follows a logic rationale and the results look promising. However, there are a few issues that the authors should address:
- The authors state that IL-9 has been show to have direct anti-tumour effect on some cell lines (lines 69-77). Can the authors tell whether the anti-tumour effect of the IL-9-armed oVV in this study is through a direct effect on the tumour or indirect, by enhancing the anti-tumour immune response? Did the authors try to treat tumour-bearing mice with IL-9 and/or unarmed oVV + IL-9 as controls?
- Have the authors measured expression of IL-9 receptor on the tumour cell lines before inoculation into the mice?
- Do the authors have data on the use of the IL-9-armed oVV in mice inoculated with cell lines knocked out for the IL-9 receptor?
- The authors state that in the tumours treated with IL-9-armed oVV, there is a reduction of MDSCs. However, in Fig 4E, the graph shows there is no significant statistical difference between the PBS vs oVV-IL-9 groups; or unarmed VV vs oVV-IL-9.
- The authors show increased expression of checkpoint molecules PD-1/PDL1 and CTLA-4 on cells from mice treated with IL-9-armed virus. Then they add anti-CTLA-4 to the IL-9-armed virus and observe a significant improvement in the treatment efficacy. However, they do not address the PD-1/PDL1 axis, which is also increased. What is the explanation for the choice of targeting CTLA-4 and not PD-1? or both CTLA-4 and PD-1?
- The authors should add a paragraph in the discussion to address the limitations of the study.
Comments on the Quality of English Language
- Minor corrections of typos.
